# Characterization of Differentially Expressed Circulating miRNAs in Metabolically Healthy versus Unhealthy Obesity

**DOI:** 10.3390/biomedicines9030321

**Published:** 2021-03-21

**Authors:** Susana Rovira-Llopis, Rubén Díaz-Rúa, Carmen Grau-del Valle, Francesca Iannantuoni, Zaida Abad-Jimenez, Neus Bosch-Sierra, Joaquín Panadero-Romero, Víctor M. Victor, Milagros Rocha, Carlos Morillas, Celia Bañuls

**Affiliations:** 1Service of Endocrinology, University Hospital Doctor Peset, Foundation for the Promotion of Health and Biomedical Research in the Valencian Region (FISABIO), 46017 Valencia, Spain; srovirallopis@gmail.com (S.R.-L.); rdiazrua@gmail.com (R.D.-R.); grau_cardel@gva.es (C.G.-d.V.); franian@alumni.uv.es (F.I.); zaiaji@alumni.uv.es (Z.A.-J.); neusboschsi@gmail.com (N.B.-S.); vmviktor@gmail.com (V.M.V.); milagros.rocha@uv.es (M.R.); carlos.morillas@uv.es (C.M.); 2Genomic Unit, La Fe Health Research Institute, 46026 Valencia, Spain; joaquin.panadero@igenomix.com; 3CIBERehd-Department of Pharmacology and Physiology, University of Valencia, 46015 Valencia, Spain; 4Department of Physiology, University of Valencia, 46015 Valencia, Spain; 5Department of Medicine, University of Valencia, 46015 Valencia, Spain

**Keywords:** obesity, metabolic syndrome, microRNAs, insulin resistance, atherogenic dyslipidaemia, oxidative stress

## Abstract

Obese individuals without metabolic comorbidities are categorized as metabolically healthy obese (MHO). MicroRNAs (miRNAs) may be implicated in MHO. This cross-sectional study explores the link between circulating miRNAs and the main components of metabolic syndrome (MetS) in the context of obesity. We also examine oxidative stress biomarkers in MHO vs. metabolically unhealthy obesity (MUO). We analysed 3536 serum miRNAs in 20 middle-aged obese individuals: 10 MHO and 10 MUO. A total of 159 miRNAs were differentially expressed, of which, 72 miRNAs (45.2%) were higher and 87 miRNAs (54.7%) were lower in the MUO group. In addition, miRNAs related to insulin signalling and lipid metabolism pathways were upregulated in the MUO group. Among these miRNAs, hsa-miR-6796-5p and hsa-miR-4697-3p, which regulate oxidative stress, showed significant correlations with glucose, triglycerides, HbA1c and HDLc. Our results provide evidence of a pattern of differentially expressed miRNAs in obesity according to MetS, and identify those related to insulin resistance and lipid metabolism pathways.

## 1. Introduction

Obesity is characterized by numerous deleterious metabolic effects and increased cardiovascular risk. However, in some individuals, these effects are negligible or not present at all, and such cases are referred to as metabolically healthy obese (MHO). There is no consensus regarding the definition of metabolically healthy obesity, but this subset of obese subjects is usually characterized by a lower waist circumference, better physical fitness, preserved insulin sensitivity, and a low prevalence of metabolic risk factors, despite their high body mass index (BMI) [1,2].

Obesity is known to be the result of environmental and genetic factors. However, only a small proportion of heritability—around 2%—is explained by the genes related to obesity, which have been determined by genome-wide association studies [3], and many of the findings regarding genes associated with high BMI have not been replicated in a reliable manner [4]. Recent studies suggest that, in addition to genetic susceptibility, epigenetics and gene–environment interaction also contribute to the appearance of obesity [5]. Unfortunately, no research has specifically set out to decipher the main genes associated with the MHO phenotype. An increasing number of studies support the involvement of epigenetics, including microRNAs (miRNAs), in different features of the metabolic syndrome [6,7,8]. Nevertheless, unravelling the molecular basis of obesity, and particularly that of the MHO population, remains an important challenge.

miRNAs are small non-coding RNA sequences of around 22 bp that regulate gene expression, and their main role is to repress the translation of specific mRNAs [9,10,11]. These molecules are involved in a multitude of physiological and pathological situations, such as obesity and metabolic disorders [12,13]. miRNAs are critical for the regulation of gene expression, and a mere few hundred are estimated to control 30–80% of the human genome, each of which has multiple target genes and is regulated by multiple miRNAs. Due to the complexity of gene–miRNA interaction, the critical role of miRNAs in MHO subjects is still poorly understood [14].

In the present study, we applied high-throughput technology to screen more than 3500 human miRNAs in obese subjects subdivided into two groups—MHO vs. metabolically unhealthy obese (MUO)—in an attempt to determine the miRNAs that differentiate these two conditions.

## 2. Materials and Methods

### 2.1. Subjects

A total of 20 patients between 18 and 70 years old with different grades of obesity (BMI > 30 kg/m^2^) diagnosed for at least five years were recruited at the Outpatient’s Clinic of the Endocrinology and Nutrition Department of University Hospital Dr. Peset in Valencia, Spain. All patients had maintained a stable weight (±2 kg) over the three months prior to the study. Patients were subdivided into two groups: metabolically healthy obese (MHO) and metabolically unhealthy obese (MUO). MUO was defined when at least two of the following features of metabolic syndrome (MetS) were present: triglyceride ≥ 150 mg/dL; HDL cholesterol (HDLc) < 40 mg/dL for men and <50 mg/dL for women, or use of lipid-lowering drugs; fasting glucose ≥ 100 mg/dL or anti-diabetic treatment; systolic (SBP) and/or diastolic (DBP) blood pressure ≥ 130/85 mmHg, respectively; or use of antihypertensive medication. Conversely, MHO was confirmed when all the aforementioned criteria for MetS were absent. Waist circumference threshold was >102 cm for men and >88 cm for women in both groups.

Exclusion criteria were secondary obesity (Cushing’s syndrome, hypothyroidism), pregnancy or lactation, severe diseases such as malignancies, renal or hepatic conditions, psychiatric disorders, drug or alcohol use, chronic inflammation, cardiovascular disease and insulin treatment.

The study was conducted according to the guidelines laid down in the Declaration of Helsinki-based Ethical Principles for Medical Research and was approved by the Ethics Committee of the University Hospital Dr. Peset (Ceic code: 35/16). Written informed consent was obtained from all subjects.

Anthropometrical parameters, including weight (kg), height (m), BMI (kg/m^2^), waist circumference (cm) and SBP and DBP (mmHg), were measured in all the participants.

### 2.2. Blood Sampling

Blood samples were obtained from the antecubital vein after 12h overnight fasting. Tubes without anticoagulant (BD Vacutainer, Franklin Lakes, NJ, USA) were stored for 30 min at room temperature and centrifuged at 1500× *g* for 10 min at 4 °C, after which the supernatant serum fraction was frozen at −80 °C for subsequent isolation of miRNA.

Biochemical determinations were carried out by the hospital’s Clinical Analysis Service. To establish lipid profile, serum levels of total cholesterol (TC) and triglycerides were measured by an enzymatic method, and HDL cholesterol (HDLc) was determined by a direct method using a Beckman LX20 analyser (Beckman Corp., Pasadena, CA, USA). The intraserial variation coefficient was <3.5% for all determinations. Levels of LDL cholesterol (LDLc) were estimated using the Friedewald equation when triglycerides were below 300 mg/dL. High-sensitive C-reactive protein (hsCRP) and apolipoprotein (Apo) AI and B levels were measured with an immunonephelometric assay (the intra-assay variation coefficient was <5.5%). Glucose and insulin were assessed by enzymatic and chemiluminescent immunoassays, respectively. Insulin resistance index was calculated using the homeostasis model of assessment (HOMA-IR = (fasting insulin (μU/mL) × fasting glucose (mg/dL)/405)). Percentage of HbA1c was assessed with an automatic glycohaemoglobin analyser (Arkray Inc., Kyoto, Japan).

### 2.3. miRNA Isolation and Microarray Analysis

Serum circulating miRNAs were isolated using the miRNeasy Serum/Plasma Kit (Qiagen, Hilden, Alemania). Affymetrix miRNA 4.0 (Santa Clara, CA, USA) was employed to study the differential expression of 3536 human miRNAs. Intensity backgrounds were corrected by RMA and normalized by quantiles. The statistical error of the repeated measured data was isolated for normalization, allowing multiple chips to be compared and analysed together.

Differential gene expression assessment of all comparisons was carried out using limma moderated t-statistics. A *t*-test statistic is reported for each gene, together with its corresponding *p*-value.

Non-mature human miRNAs were excluded from downstream analyses. miRTarBase 7.0 (http://miRTarBase.cuhk.edu.cn/, accessed on 11 March 2021) [15] was employed to search for experimentally validated target genes for the most up- and downregulated miRNAs. If none was found in this way, target prediction was based on TargetScanHuman 7.2 (http://www.targetscan.org/vert_72/, accessed on 11 March 2021) [16]. Biological pathway assignment was performed manually using GO, KEGG, Biocarta and WikiPathways (https://www.wikipathways.org, accessed on 11 March 2021) databases.

Heatmaps of correlation distance and average linkage of miRNA expression levels were created in MUO and MHO subjects using the web tool Clustvis (https://bio.tools/clustvis, accessed on 11 March 2021) [17].

Venn diagram was constructed with the on-line resource http://bioinformatics.psb.ugent.be (last accessed on 11 March 2021).

### 2.4. miRNA Target Prediction and Encyclopaedia of Genes and Genomes (KEGG) Pathway Analyses

In silico target prediction of miRNA was performed using the miRBD database [18]. We selected the datasets by considering a target score value > 80. KEGG pathway analyses were carried out with GlueGo+CluePediaCytoscape plugin [19,20].

### 2.5. miRNAs qPCR Validation

Three upregulated miRNAs from the screening study were chosen for validation based on the following criteria: levels of expression in the microarray, and biological and statistical significance.

For validation, RNA was converted into cDNA using the TaqMan Advanced miRNA cDNA Synthesis Kit (Thermo Fisher Scientific, Waltham, MA, USA) following the manufacturer’s instructions. The expression levels of hsa-mir4697-3p, hsa-mir-588, hsa-mir-6796-5p and the spike-in control cel-miR-39 were assessed by qPCR with specific Taqman hydrolysis probes (Thermo Fisher Scientific) in a 7500 Fast Real-Time PCR System (Applied Biosystems, Foster City, CA, USA). Relative gene expression was obtained using the 2−ΔΔCT method with ExpressionSuite software (Applied Biosystems).

### 2.6. Data Analysis

Statistical analysis was performed with the SPSS program version 17.0 (last accessed on 28 December 2020). Data are expressed as the mean and standard deviation in tables and standard error of the mean (SEM) in figures. An unpaired student’s *t*-test was used to compare variables between MHO vs.MUO groups. Bivariate correlation amongthe expression levels of individual miRNA and each ofthe anthropometric and biochemical determinations wasmeasured using Spearman’s correlation coefficient by JMP PRO 14.1. software, and a heatmap was generated with GraphPad Prism version 7.0 for Windows (GraphPad Software, San Diego, CA, USA, www.graphpad.com, last accessed on 28 December 2020). All differences were considered significant when *p* < 0.05. 

## 3. Results

### 3.1. Characteristics of the Study Cohort

A total of 10 MHO and 10 MUO subjects were analysed. MUO subjects presented higher BMI, waist circumference and blood pressure than their MHO counterparts (*p* < 0.05 in all cases; Table 1). Lipid profile showed typical features of atherogenic dyslipidaemia in MUO subjects, with higher triglyceride levels and lower HDLc levels than in the MHO group (*p* < 0.001). ApoA1 levels were lower in MUO subjects (*p* < 0.01) and Apo B levels were higher in comparison to MHO (*p* < 0.05). As expected, the MUO group displayed higher glucose (*p* < 0.001), HbA1c (*p* < 0.01), insulin and HOMA-IR levels (*p* < 0.01), indicating hyperglycaemia and insulin resistance. hsCRP levels were similar in both groups.

### 3.2. Evaluation of Differentially Expressed miRNA

Our data indicated that, among the 3536 miRNAs studied, 159 were differentially expressed between the two groups, of which 72 were upregulated and 87 were downregulated in the serum of MUO versus MHO subjects (Figure 1; (dark dots) *p*-value < 0.05). Table 2 shows the top 5 up- and downregulated miRNAs and their predicted target genes.

### 3.3. Heatmap

This heatmap indicates that MUO and MHO subjects could be differentiated by the expression of dysregulated miRNAs (Figure 2). The vertical dendrogram shows a clustering in MUO vs. MHO patients.

### 3.4. Pathway Enrichment

Differential miRNAs in MUO vs. MHO were manually assigned to different biological processes, and it was observed that most processes corresponded with a similar percentage of up- and downregulated miRNAs (Figure 3A). A more pronounced upregulation of miRNAs was related to several pathways in the MUO group, specifically insulin signalling and lipid metabolism pathways.

### 3.5. In Silico Analysis of Potential miRNA Target Genes

To explore the potential roles of the top ten upregulated miRNAs, we performed a target gene analysis using the bioinformatics analysis tool miRDB. These algorithms generated a total of 4389 predicted affected genes for the top ten upregulated miRNAs. We selected target genes with a target score >80 to assure higher confidence in the prediction (total predicted genes 1.177). After removing duplicates and triplicates, we submitted a final list of 1.104 predicted target genes to KEGG pathway analysis using a GlueGo+CluePediaCytoscape plugin (cytoscape.org, accessed on 11 March 2020). The genes predicted to be targets of the miRNAs were significantly enriched for 53 terms that constituted five categories of functional networks: regulation of metabolic processes, neurogenesis, regulation of primary metabolic processes, positive regulation of cellular process and cellular response to endogenous stimulus. The results are shown in Figure 3B,C and detailed in Appendix A.

### 3.6. Validation of a Subgroup of miRNAs

Since we had observed high ROS production in MUO in a previous study [21], among the MUO-upregulated miRNAs involved in insulin signalling and lipid metabolism pathways we identified those that are also associated with modulation of oxidative stress. In this way, three miRNAs were selected for further validation by RT-PCR: hsa-miR-588, hsa-miR-6796-5p and hsa-miR-4697-3p (red dots in Figure 1). Their normalized expression in the microarray is shown in Figure 4A. We subsequently confirmed that hsa-miR-6796-5p and hsa-miR-4697-3p were upregulated in the MUO vs. MHO group (*p* < 0.05) (Figure 4B–D).Experimentally validated targets of these 3 miRNAs are included in Appendix A. No significant correlations were found among the expression of these 3 miRNAs (data not shown).

Figure 5A shows that a total of 40, 10, 28 and 6 processes related to lipid metabolism, oxidative stress, insulin signalling and glucose homeostasis are target ofthese three upregulated miRNAs (complete list of processes in Appendix A). Interestingly, the Venn diagram (Figure 5B) shows three processes that were commonly affected (R-HSA-1430728_metabolism; R-HSA-1257604_PIP3 activates AKT signalling; and R-HSA-556833_metabolism of lipids) by the validated miRNAs. It could suggest a strong regulation of these metabolic pathways.

### 3.7. Correlation Studies

Among the three validated miRNAs, hsa-miR-4697-3p was positively correlated with glucose and triglycerides and negatively correlated with HDLc (*p* < 0.05). A positive correlation was observed between hsa-miR-6796-5p and insulin (*p* < 0.01), glucose, triglycerides, A1c, BMI and waist circumference (*p* < 0.05 for all). However, in the case of hs-miR-588, we detected a significant negative correlation only with insulin (*p* < 0.01) (Figure 6).

In terms of the most significant correlations observed, the biochemical parameters of metabolic function that showed the closest association with the different up- and downregulated miRNAs were glucose, triglycerides, HbA1c, and HDLc (Figure 6).

## 4. Discussion

This study provides a detailed profile of serum-circulating miRNAs associated with obesity according to metabolic condition. As expected, a remarkable alteration was observed in the biochemical metabolic profile of our MUO population, and this profile was accompanied by substantial differences in the pattern of miRNA expression, including 72 upregulated and 87 downregulated miRNAs in the serum of MUO vs.MHO subjects. In addition, an upregulation of miRNAs related to insulin signalling and lipid metabolism pathways was evident in the MUO group. Two miRNAs involved in the insulin signalling pathway and also related to oxidative stress—hsa-miR-6796-5p and hsa-miR-4697-3p—were upregulated in the same group. Correlation studies revealed a relationship between miRNA expression and markers of metabolic profile in the obese population; namely, insulin, glucose and triglycerides.

A particular subset of obese patients without metabolic alterations has been categorized within the concept of MHO; however, there are not well-established diagnostic criteria for this classification. In this sense, miRNA studies could provide new insight into the relevance of miRNA regulation in the development of obesity-related metabolic disorders. Among the 159 miRNAs that showed significant differences between MUO and MHO subjects, circulating levels of 72 miRNAs (45.2%) were higher and those of 87 miRNAs (54.7%) were lower in the MUO group. These account for nearly 5% of the miRNA species profiled in the assay, which suggests that the overall miRNA balance is disrupted more in MUO than in MHO. The noticeable difference in expressed miRNAs reflects a significant change in miRNA-dependent intercellular signalling [9]. In line with these results, Choy et al. reported that the levels of 65 of the 374 circulating miRNAs they had profiled were elevated in insulin-resistant obese subjects, and miR-192-5p, miR-194-5p, miR-486-5p, miR-150, miR-378a-3p, miR-550a-3p, miR-16-5p, and miR-140-3p have previously been reported to be associated with insulin resistance [22].

In the present study, the non-targeted profiling of circulating miRNAs and predicted functional pathways points to the involvement of several of these miRNAs in the regulation of cell growth, differentiation and cell adhesion. Our initial pathway analysis revealed similar numbers of up- and downregulated genes for a given pathway. This observation, and the fact that miRNAs can target genes that either positively or negatively regulate the same pathway, adds complexity to our analysis. We have focused on pathways in which upregulated miRNAs predominate over downregulated ones. Interestingly, miRNAs related to insulin signalling and lipid metabolism were particularly pronounced in the MUO group.

Several miRNAs are reported to play regulatory roles in many biologic processes related with obesity, including insulin resistance, lipid metabolism and adipocyte differentiation [13,23,24]. Doumatey et al. have recently shown that circulating miR-374a-5p is upregulated in MHO vs.MUO subjects and is associated with the TG/HDL ratio. In addition, this miRNA may modulate the expression of CCL2, a pro-inflammatory biomarker found upstream in the pathway that leads to dyslipidaemia in obesity. Therefore, by regulating CCL2 expression, miR-347a-5p may modulate inflammation and alter lipid profile in obesity [25]. Interestingly, miR-126, one of the top downregulated miRNAs in this study, has previously been related to the regulation of adipose tissue inflammation in human obesity, since it also targets CCL2, inhibiting its expression and release from adipose tissue [26]. Thus, the downregulation of miR-126 in MUO patients may contribute to their underlying inflammatory state, partly due to a lack of repression of CCL2 production in adipose tissue.

Obesity, insulin resistance and dyslipidaemia have a strong connection with type 2 diabetes, and several miRNAs influence these clinical outcomes. When we analysed the miRNAs that were differentially expressed in obese patients, miR-6822-3p, miR-23a, miR-4439, miR-512-5p, miR-6796-5p, miR-8088, miR-3622b, and miR-4532 were found to be significantly correlated with metabolic parameters associated with insulin resistance. In this context, Ma et al. reported that levels of circulating miR-150 and miR-16-3p are correlated with the insulin sensitivity index when measured by the euglycemic clamp experiment [27], and miR-378a has been shown to be directly involved in energy and fat metabolism [28]. Several of these upregulated circulating miRNAs, including miR-150-5p, 16-5p, 192-5p, 451a, 486-5p, and 770-5p, are reported to be expressed in patients with type 2 diabetes [29].

Oxidative stress and inflammation are related to obesity, but the effect of metabolic disturbances on these parameters and their relationship with miRNAs is unknown. In obese and overweight individuals MetS displays differences in terms of chronic inflammation, nitro-oxidative stress and insulin resistance [21,30,31,32]. Several miRNAs seem to be associated with the downregulation of pro-inflammatory markers linked to insulin resistance. For instance, miR-126, miR-132, miR-146, miR-155, and miR-221 have emerged as important transcriptional regulators of some inflammation-related mediators [33]. The two miRNAs we identified in the MUO group—and which were validated by qPCR, miR-6796-5p and miR-4697-3p—were assigned prostaglandin-endoperoxide synthase (COX-2) and glutathione peroxidase 7 (GPX7) as predicted target genes, respectively. These are key enzymes in the generation and scavenging of reactive oxygen species; for this reason, these miRNAs should be explored in more detail, as they seem to be implicated in the metabolic status of obesity.

It is known that metabolic disorders and/or obesity are a high-risk factor for carcinogenesis. In this context, we identified important tumour suppressors or oncomiRs among our top five up- and down- regulated miRNAs. miR-548 has been implicated in the regulation of gene expression related to endometriosis and ovarian cancer [34]. In this context, several tumour suppressor pathways act on or are activated by the signal transducer and activator of transcription (STAT)3 signalling, including miR-1181 [35]. Additionally, miR-126 inactivates the PI3K/AKT pathway in the carcinogenesis process [36]. Increasing evidence supports that, as a new tumour suppressor gene, miR-126 plays a role in the development and metastasis of various types of cancer, such as lung cancer, liver cancer, melanoma and colorectal cancer [37].

Moreover, a critical role of miR-137 has previously been described in the pathophysiology of cancer, specifically in glioma [38]. Conversely, downregulation of miR-137 ameliorates high glucose-induced injury in HUVECs by overexpression of AMPKα1, leading to increased cellular reductive reactions and decreased oxidative stress [39].

Regarding the rest of our top modulated miRNAs, no studies have implicated miR-4532, miR-6798-3p, miR-3157-39, miR-5047, miR-4508 or miR-5090 in human disease.

A strength of this study is that we selected two phenotypes of obese individuals clearly differentiated according to the presence of three or four comorbidities associated with MetS. In addition, the subjects were of a similar age and both sexes were represented equally.

On the other hand, we should point out some limitations to our work. First of all, it is possible that we have overlooked the expression of relevant miRNAs due to the small sample size of this non-targeted profiling study. However, qPCR validation of the 3 selected miRNAs further supported their upregulation in the MUO condition. In addition, the relationship between these circulating miRNAs and metabolic alterations is supported by the roles of many of their target genes in obesity and lipid metabolism (Figure 5; Appendix A). In this sense, our findings need to be confirmed in specific populations in order to exclude potential bias. Finally, our analysis of target pathways and genes was limited to prediction analysis; consequently, future studies addressing the mechanisms of action of the miRNAs we have identified are necessary before our findings can be translated to clinical practice.

In summary, our data suggest that the miRNAs that regulate insulin signalling, lipid profile and oxidative stress, and which we have identified in our population, play an essential role in protecting against metabolic disease in obesity. These miRNAs are also correlated with clinical and biochemical markers of metabolic dysfunction—chiefly, insulin resistance. Our findings encourage the targeted research of serum miRNAs as clinically relevant biomarkers in the treatment of metabolic disease in obesity. Furthermore, our study points to serum miRNA profiling as a promising tool with which to accurately classify obese subjects as MHO or MUO, thus improving the diagnosis and early treatment of these patients.

## Figures and Tables

**Figure 1 biomedicines-09-00321-f001:**
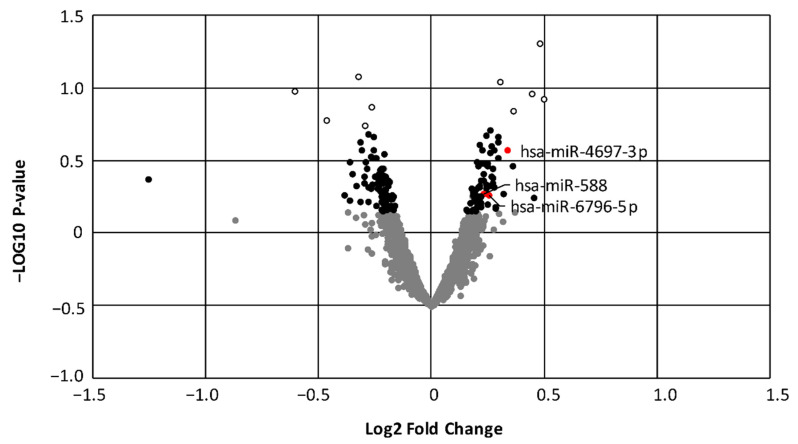
Volcano plot representing differentially expressed miRNA in MHO and MUO groups. Black dots indicate significantly different levels of miRNAs in MUO vs. MHO. Red dots are the 3 miRNAs chosen for validation. Empty dots are the top 5 up- and downregulated miRNAs.

**Figure 2 biomedicines-09-00321-f002:**
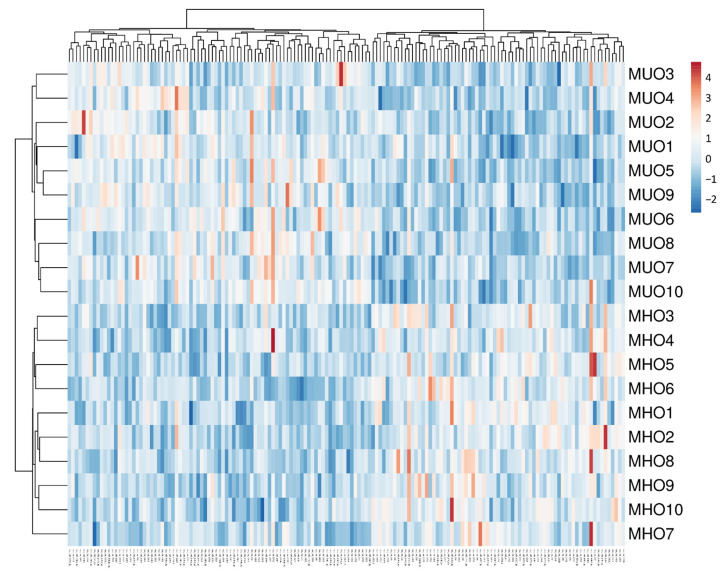
Hierarchical clustering of samples (MHO and MUO, *n*=10 in each group) based on summarized intensity values of the 159 differentially expressed circulating miRNAs. Log2 intensity values are shown in the bar scale. Rows are centred; unit variance scaling has been applied to rows. Both rows and columns are clustered using correlation distance and average linkage. 20 rows, 156 columns. Annotations on the heatmap show clustering of the samples.

**Figure 3 biomedicines-09-00321-f003:**
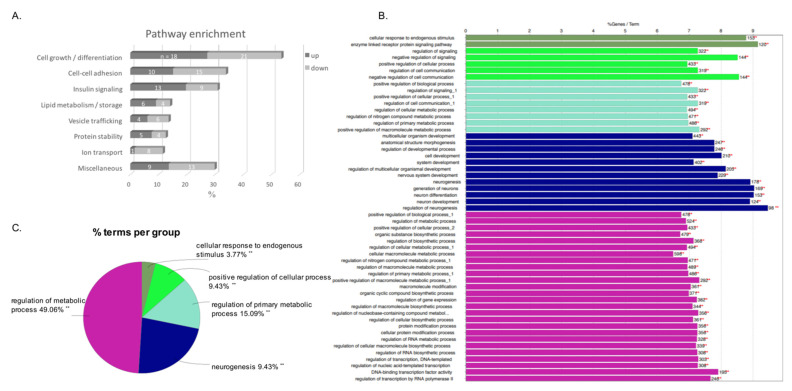
(**A**) Top 40 biological processes and pathways associated with up- and downregulated miRNAs in MUO individuals. (**B**) Upregulated GO/pathway terms specific for target genes. Bars represent the number of genes associated with the terms. The percentage of genes per term is labelled in the bar. (**C**) Overview chart with the enriched functional groups including specific terms for target genes.

**Figure 4 biomedicines-09-00321-f004:**
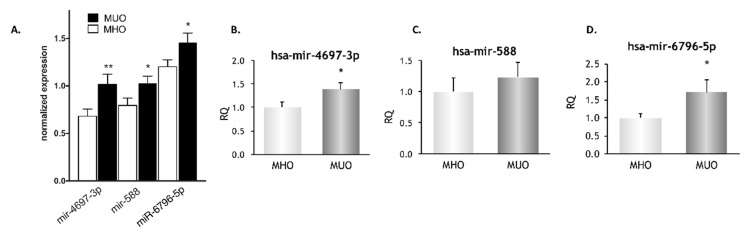
Validation of 3 miRNA array-predicted changes by quantitative RT-PCR (qPCR). (**A**) Microarray expression of the three selected miRNAs in MHO and MUO subjects. (**B**–**D**) Quantitative miRNA expression analysis of (**A**) hsa-mir-4697-3p (**B**) hsa-mir-588 and (**C**) hsa-mir-6796-5p was carried out using TaqMan probes by qPCR. * *p* < 0.05, ** *p* < 0.01.

**Figure 5 biomedicines-09-00321-f005:**
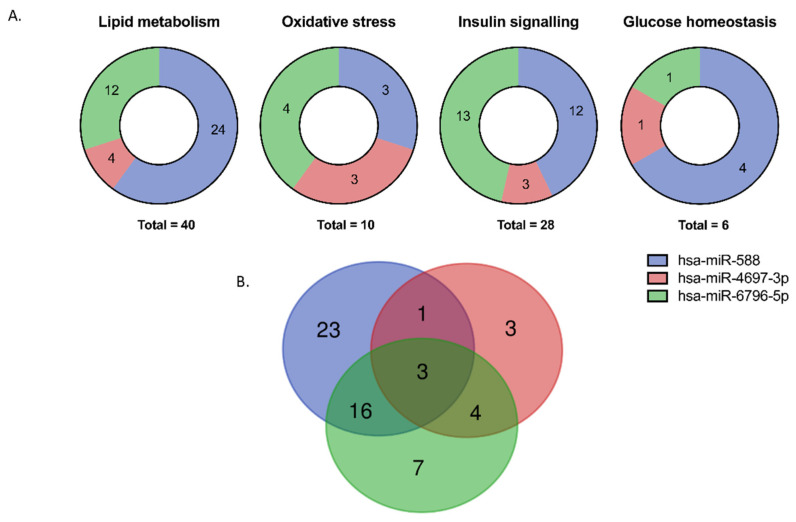
(**A**) Pie charts showing the numbers of affected metabolic processes by each validated miRNA in each main pathway (lipid metabolism, oxidative stress, insulin signalling, and glucose homeostasis). (**B**) Venn diagram of unique and shared processes by each validated miRNA. The different miRNAs are coded by colour. Analysis performed at https://reactome.org, (accessed on 11 March 2020) with the target genes for each miRNA according to miRcarta v1.1 (details in Appendix A).

**Figure 6 biomedicines-09-00321-f006:**
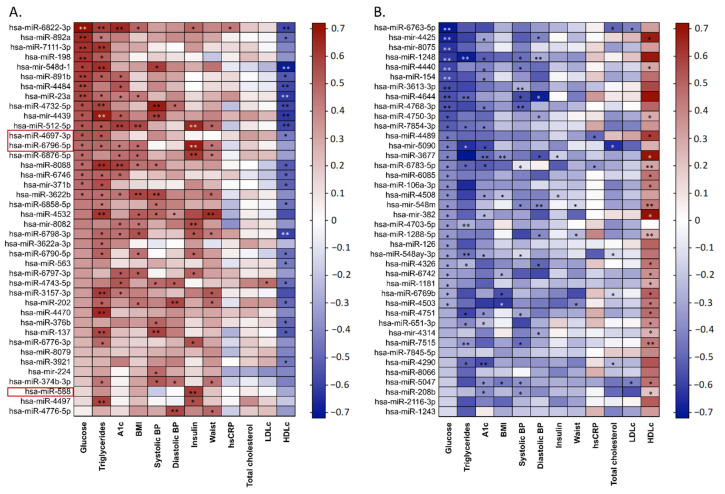
Correlations between top 40 upregulated (**A**) and downregulated (**B**) miRNAs and anthropometric and biochemical parameters in the MUO group. * *p* < 0.05; ** *p* < 0.01. Colour keyrepresents Spearman rank correlation coefficients in a colour gradient (red: positive and blue negative values).

**Table 1 biomedicines-09-00321-t001:** Clinical and metabolic variables in metabolically healthy obese and metabolically unhealthy obese individuals.

	MHO	MUO	*p*-Value
*n* (% men)	10 (20)	10 (30)	ns
Age (years)	44.2 ± 10.5	45.8 ± 6.5	ns
BMI (kg/m^2^)	38.1 ± 2.86	42.9 ± 4.37	<0.05
Waist (cm)	115.3 ± 11.6	129.4 ± 11.8	<0.05
Systolic BP (mm Hg)	122 ± 10	142 ± 14	<0.01
Diastolic BP (mm Hg)	76 ± 6	89 ± 11	<0.05
Total cholesterol (mg/dL)	178 ± 29	194 ± 25	ns
LDLc (mg/dL)	108 ± 23	115 ± 21	ns
HDLc (mg/dL)	55 ± 7.7	35 ± 3.3	<0.001
Triglycerides (mg/dL)	70.5 (47.0–102.0)	225 (186–244)	<0.001
Apo AI (mg/dL)	164 ± 18	134 ± 15	<0.01
Apo B (mg/dL)	89 ± 20	114 ± 19	0.05
Glucose (mg/dL)	87.1 ± 8.4	107.2 ± 8.1	<0.001
Insulin (μU/mL)	13.5 ± 6.9	27.3 ± 16.4	<0.05
HOMA-IR	2.85 ± 1.29	7.16 ± 4.12	<0.01
HbA1c (%)	5.22 ± 0.37	5.91 ± 0.40	<0.01
hsCRP (mg/L)	4.57 (2.83–11.17)	5.33 (2.27–11.78)	ns

MHO: metabolically healthy obese; MUO: metabolically unhealthy obese; BMI:body mass index; BP: blood pressure; LDLc: low density lipoprotein cholesterol; HDLc: high density lipoprotein cholesterol; Apo: apolipoprotein; HOMA-IR: homeostasis model of assessment of insulin resistance index; HbA1c: glycosylated haemoglobin; hsCRP: High-sensitive C-reactive protein; ns: non-significant.

**Table 2 biomedicines-09-00321-t002:** Top 5 up- and downregulated miRNAs in MUO vs. MHO subjects.

Direction	Name	Target Gene	Fold Change	*p*-Value
Up	hsa-miR-4532	POU3F1	1.39	0.0002
	hsa-miR-548d-1	PPARA *	1.23	0.0008
	hsa-miR-3157-3p	OR11A1	1.36	0.0012
	hsa-miR-137	CDK6 *	1.41	0.0014
	hsa-miR-6798-3p	PEBP1	1.29	0.0020
Down	hsa-miR-5047	TFRC	0.79	0.0007
	hsa-miR-4508	BOK	0.66	0.0011
	hsa-miR-1181	STAT3 *	0.83	0.0018
	hsa-miR-126	CRK *	0.72	0.0027
	hsa-miR-5090	WNK2	0.81	0.0033

* Experimentally validated target (by reporter assay/Western blot/qPCR) [15,16].

## Data Availability

The data presented in this study have been deposited in NCBI's Gene Expression Omnibus (Rovira-Llopis S et al., 2021) are openly available are accessible through GEO Series accession number GSE169290 (https://www.ncbi.nlm.nih.gov/geo/query/acc.cgi?acc=GSE169290 (accessed on 22 March 2021)).

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
