# Peer review of "Characterization of Differentially Expressed Circulating miRNAs in Metabolically Healthy versus Unhealthy Obesity"

_biomedicines, 2021, doi:10.3390/biomedicines9030321_

Round 1

Reviewer 1 Report

In this manuscript, the authors investigated serum containing miRNAs profile between metabolically healthy obese (MHO) and metabolically unhealthy obesity (MUO), and found several important biomarkers, involved in insulin signaling and oxidative stress, e.t.c. Overall the manuscript re well written and interesting. The authors should consider the moderate concerns as below to further improve the manuscript before it becomes publication for this journal.

  1. Include miRNA expression intensity graph and Pearson's product moment correlation coefficient between MUO and MHO, in addition to volcano plot in Figure1. Because the expression of miRNA species varies among tissue, cell types, something like that. Especially, my concerns are absolute expression level of miRNAs that the author picked up.

  1. In Figure 2, miss-labeled MAO?

  1. Figure 3A shows similar percentage of biological processes between up or down-regulated miRNAs. What it mean? Any data for pathway analysis using only up-or down regulated miRNAs?

  1. Figure 4B shows non-significant change? Any other validation test?

  1. Figure 5: Include the detail about r value in Method.

Reviewer 2 Report

This is a small paper that aims to highlight the deregulated miRs between healthy and unhealthy obesity. The subject is highly interesting however there are so many undelivered promises at the current version of the paper. I appreciate that authors used a small group of patient samples, n=20 however their results are not convincing.

  1. In the introduction the authors named specific SNPs, what is the relevance with the current study? The discussion and the results are not relevant with the intro. Any info about the current cohort and their SNPs? If not what is the point of mentioning these SNPs?
  2. Results, table 2 summarises the fold changes for top miRs, none of the fold changes are more than 2 folds. I am not convinced about their significance at this stage. qPCR data is there but the changes in expression levels are very small. WB and the reporter assay results are needed to be included. The pathway analysis is great but the current data needs to be stronger.
  3. Discussion, focuses on more miRs that they were not in the top 5 up/down regulated list. Some of these miRs are highly important tumour suppressors or oncomiRs, unfortunately once again authors do not address these. Metabolic disorder and/or obesity is a high risk for carcinogenesis therefore these miRs should be discussed properly.

Round 2

Reviewer 2 Report

The manuscript is improved. I suggest authors to include a diagram to summarise the miRNAs & signalling pathways in the discussion as the supplementary data not always seen. 

Author Response

We appreciate the reviewer's suggestion. We have accordingly added a new figure (Figure 5), representing affected metabolic processes by each validated miRNA and those pathways that are shared among the 3 selected miRNAs.